# On Neural Scaling Laws for Weather Emulation through Continual Training

**Shashank Subramanian**[1][*] **Alexander Kiefer**[2], **Arnur Nigmetov**[1],
**Amir Gholami**[3,4], **Dmitriy Morozov**[1], **Michael W. Mahoney**[1,3,4]

[1]Lawrence Berkeley National Laboratory    [2]Oak Ridge National Laboratory
[3]International Computer Science Institute    [4]University of California at Berkeley

## Abstract

Neural scaling laws, which in some domains can predict the performance of large neural networks as a function of model, data, and compute scale, are the cornerstone of building foundation models in Natural Language Processing and Computer Vision. We study neural scaling in Scientific Machine Learning, focusing on models for weather forecasting. To analyze scaling behavior in as simple a setting as possible, we adopt a minimal, scalable, general-purpose `Swin` Transformer architecture, and we use continual training with constant learning rates and periodic cooldowns as an efficient training strategy. We show that models trained in this minimalist way follow predictable scaling trends and even outperform standard cosine learning rate schedules. Cooldown phases can be re-purposed to improve downstream performance, e.g., enabling accurate multi-step rollouts over longer forecast horizons as well as sharper predictions through spectral loss adjustments. We also systematically explore a wide range of model and dataset sizes under various compute budgets to construct IsoFLOP curves, and we identify compute-optimal training regimes. Extrapolating these trends to larger scales highlights potential performance limits, demonstrating that neural scaling can serve as an important diagnostic for efficient resource allocation. We open-source our code for reproducibility.

## 1 Introduction

Machine learning models, and deep learning models in particular, have demonstrated great success in scientific problems, including emulating atmospheric physics for weather forecasting. In particular, in recent years, several data-driven models (Lam et al., 2023; Bonev et al., 2025; Bodnar et al., 2024; Bi et al., 2022; Ben-Bouallegue et al., 2023; Price et al., 2023; Alet et al., 2025) have matched or surpassed the accuracy of gold-standard Numerical Weather Prediction (NWP) systems that generate high-fidelity forecasts by solving complex, multiscale fluid-dynamics equations. These data-driven models can generate weather forecasts orders of magnitude faster than classical NWP models using substantially lower resources, often only a single GPU, in inference.

The training costs of these deep learning weather emulators are rapidly increasing, as researchers explore a wide range of architectural choices, loss functions, and training methodologies in pursuit of state-of-the-art forecasting performance. These costs are further amplified by the growing scale of models, reaching $O(100)$ billion parameters, as well as increasing data resolutions, e.g., through smaller patch sizes in Transformer-based architectures (Hatanpää et al., 2025; Wang et al., 2024; 2025). Similar scaling challenges arise in areas such as Natural Language Processing (NLP) and Computer Vision (CV). However, scientific data present additional unique challenges, perhaps most notably their spatiotemporal structure. The proliferation of increasingly-complex architectures, in particular in Scientific Machine Learning (SciML), raises important questions. Are we learning about weather forecasting, or science more generally, or merely about the idiosyncrasies of ever more elaborate (domain-motivated) neural architecture designs? What is the simplest possible architecture to obtain scaling that is good for downstream SciML applications?

---

[*]Correspondence to: shashanksubramanian@lbl.gov

These questions are timely since, in many areas of machine learning, progress has ultimately been driven not by increased architecture complexity, but by scale—more data, larger models, and more compute—a phenomenon sometimes referred to as the *bitter lesson*. Understanding how performance scales with these quantities therefore requires studying simple, general-purpose models under controlled scaling regimes. Without such a foundation, it becomes difficult to disentangle genuine scaling behavior from artifacts introduced by specialized architectural choices.

In domains such as NLP, this challenge has been addressed through the development of *neural scaling laws* (Hestness et al., 2017; Kaplan et al., 2020; Hoffmann et al., 2022). Distinct from other scaling concepts, such as strong scaling and weak scaling in high performance computing, neural scaling laws are a set of empirical patterns that guide practitioners in scaling data quantity, model size, and the amount of compute, so that none of them saturate. When this happens, the model loss follows a power-law relationship as a function of the size of the model, of the dataset, and of the computational resources available for training. Within NLP, these trends can span more than seven orders of magnitude (Hoffmann et al., 2022; Grattafiori et al., 2024). The study of neural scaling laws is important because it builds confidence and trust that model performance will improve with increased scale. This predictive aspect is valuable because it makes neural scaling laws not only an analytical method but also actionable for planning efficient resource allocation.

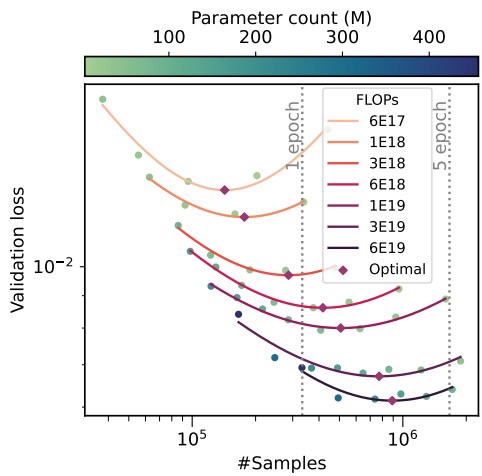

Figure 1: **Neural scaling for weather emulation.** We pre-train several models using continual training (constant learning rates with periodic cooldowns; see §3), and we identify compute-optimal regimes to train the neural emulator so that neither data nor model size saturate at different compute budgets. At each FLOP budget, several model sizes (up to 450M) are trained to different dataset sizes to form IsoFLOPs that demonstrate the tradeoff between model and data size. Unlike NLP, these models are trained for multiple epochs (indicated by vertical dotted lines), causing samples to be revisited after the first epoch and effectively be treated as pseudo-samples. We fit parabolas to each IsoFLOP, and we track the compute-optimal model at each budget.

In the existing literature, there has been limited effort in understanding the neural scaling behavior in SciML, in particular for deep learning models for weather forecasting. Existing studies do not explicitly investigate the joint relationship between model and data size, as a function of available computational budget, and/or their analyses remain incomplete. For example, Bodnar et al. (2024) reports predictable scaling with model size; but it does not perform compute-optimal analyses, it relies on scaling with GPU time rather than FLOPs, and it incorporates data outside the traditional weather-forecasting domain, such as pollutants and sea state, which makes interpreting the scaling behavior harder. In contrast, Karlbauer et al. (2024) suggests that performance stalls, for neural operators and related architectures, at even small model scales, although it should be noted that this conclusion is drawn from experiments at coarser spatial resolution than we consider. More broadly, prior works stop short of identifying compute-optimal combinations of model size and data volume at each scale, making it difficult to disentangle true scaling laws from artifacts of architectural, data, or compute choices.

In this work, we aim to systematically characterize the neural scaling relationships governing data-driven weather forecasting models. Our main contributions are:

1. **Minimalist transformer architecture for neural scaling.** Rather than designing specialized architectures for weather forecasting, we intentionally adopt a minimalist and widely used backbone, the `Swin` Transformer (Liu et al., 2021), without domain-specific architectural modifications or custom loss functions, during pre-training. This design choice follows the principle that understanding scaling behavior requires reducing architectural

confounders. Our goal is therefore not to engineer the most specialized weather model, but to understand how far simple architectures can be pushed through scale. To enable such experiments across a wide range of scales, we further implement $2D$ spatial parallelism along with data parallelism, with the former being crucial for high resolution inputs.

2. **Continual training for weather emulators.** Inspired by Hägele et al. (2024), we apply learning rate (LR) cooldowns after a period of continual training with a constant LR. Here, "continual training" refers to the ability to continue optimizing a model across different compute budgets from a checkpoint without restarting from scratch, rather than the standard multi-task continual learning setting. We show that this fixed LR followed by a cooldown protocol can outperform the standard cosine schedule commonly used in pre-training weather emulators (Bodnar et al., 2024; Lam et al., 2023) (see Fig. 2). This approach enables efficient exploration of a wide range of compute budgets without retraining models from scratch, by periodically cooling down the LR to match the desired budget. We show that even a cooldown period as short as 5% of the total training iterations is sufficient to achieve consistent performance (see Fig. 3).

3. **Re-purposing cooldowns for better downstream performance.** We demonstrate that the short cooldown period can be re-purposed with alternate loss functions to better align the pre-trained model to downstream performance. For example, we can perform multi-step rollouts in training during cooldown to achieve higher accuracy over longer forecast horizons (see Fig. 3, 4). We may also spectrally adjust the loss function (Fig. 4b) to allow for sharper forecasts to capture critical high resolution features. Importantly, this allows us to disentangle neural scaling from downstream performance, without which the scaling would be repeated for each loss function.

4. **Neural scaling to identify compute-optimal training regimes.** We construct scaling laws by pre-training several models (up to $450M$ parameters) using compute budgets from 6E+17 to 6E+19 FLOPs on the $0.25°$ ERA5 (Hersbach et al., 2020) dataset ($O(300,000)$ samples at an hourly temporal resolution). For each budget, different models are cooled down to different iterations that give rise to IsoFLOP curves (see Fig. 1, 5), a set of training configurations (model sizes and training iterations) that achieve a fixed total training FLOPs. Across these budgets, we observe clear compute-optimal scaling behavior, where each budget admits an optimal combination of model size and effective dataset size. To probe the limits of these trends, we extrapolate the scaling laws to 2.25E+21 FLOPs and train a 1.3B parameter model. This large-scale experiment shows signs of saturation before reaching the projected loss. This suggests that scaling in this regime may become limited by data size and spatiotemporal resolution, highlighting the importance of neural scaling analyses for diagnosing progress.

## 2 RELATED WORKS

**Neural weather emulators.** Models such as FourCastNet (Kurth et al., 2023) approached the accuracy of NWP in medium-range weather forecasting (10–14 days), while being orders of magnitude faster. Newer models (Bi et al., 2022; Lam et al., 2023; Bodnar et al., 2024; Bonev et al., 2025; Price et al., 2023) have surpassed the accuracy of NWP, while retaining the speedups. The architecture landscape of these models includes Neural Operators(Kurth et al., 2023; Bonev et al., 2025), graph-based models (Lam et al., 2023; Price et al., 2023), and Transformer-based models (Chen et al., 2023; Nguyen et al., 2024b; Bi et al., 2022). Willard et al. (2024) demonstrated that off-the-shelf Transformers that are trained well may be sufficient for good performance. In addition to a variety of architectures, a variety of loss functions, including standard MSE, autoregressive rollout MSE (Lam et al., 2023; Brandstetter et al., 2023; Bodnar et al., 2024), and spectral-based losses (Subich et al., 2025), have been employed, depending on the downstream objective. Despite this progress, with a focus on architectural design and accuracy, there has been relatively little effort in systematic analysis of how model design, data scale, and compute budget interact in weather emulators, which are central goals here.

**Neural scaling laws in large ML models.** Research in other domains, most notably in NLP, has established the utility of neural scaling laws for guiding model development. Kaplan et al. (2020) demonstrated that language model loss scales predictably as these axes increase, enabling researchers to estimate performance for larger models before training. Hoffmann et al. (2022) then

formalized compute-optimal training ("Chinchilla" scaling), showing how to better balance model parameters and training data. More recent efforts have refined these laws, extended them to new architectures, and explored their theoretical foundations (Grattafiori et al., 2024; Bahri et al., 2024; DeepSeek-AI et al., 2024; Krajewski et al., 2024; Abnar et al., 2025). Finally, Hägele et al. (2024) show that LR scheduler changes that allow for continual training make the cost and design of neural scaling experiments more manageable. Our methodology is inspired by this.

**Neural scaling laws in large SciML models.** There are also emerging efforts to apply scaling law concepts to SciML models, beyond NLP. For example, universal models of atomic systems have been developed with empirical scaling laws to guide capacity and data allocation across compute budgets, illustrating how scaling insights can transfer to scientific domains (Wood et al., 2025; Wadell et al., 2025). Another example in Nguyen et al. (2024a) builds foundation models that predict the function and structure of biological sequences and discover the design space through neural scaling laws. Comprehensive neural scaling studies remain scarce in weather forecasting, and existing analyses mostly do not identify compute-optimal regimes and exhibit mixed findings (Bodnar et al., 2024; Karlbauer et al., 2024). Today, large models (with billions of parameters) (Hatanpää et al., 2025; Wang et al., 2024) have already been trained for weather forecasting, further emphasizing the need to understand this design space. To the best of our knowledge, the very recent work of Yu et al. (2026) (which has appeared concurrent to this work) is the first to consider neural scaling laws systematically for weather forecasting. In Yu et al. (2026), the authors scale up to 1E+18 FLOPs for existing models in the literature, and they demonstrate how some models show flavors of compute optimal scaling, whilst others do not. The main differences in our work is the emphasis on (i) continuous training with cooldowns for efficient neural scaling and downstream alignment, (ii) scaling to over an order of magnitude higher compute (6E+19 FLOPs) via spatial parallelism to address memory constraints, (iii) demonstrating clear compute-optimal regimes through a minimalist Transformer backbone that matches state-of-the-art performance under the metrics of Rasp et al. (2024), and (iv) probing scaling limits under multi-epoch training by extending compute to O(2E+21) FLOPs. Our works aim to fill current gaps in scaling research by adapting neural scaling methodologies from NLP research to the context of data-driven weather emulation, providing systematic guidance for compute-optimal model design.

## 3 OVERVIEW OF METHODS

Neural scaling involves pre-training a series of models across a practical range of compute budgets. At each compute budget, several model sizes are trained to different numbers of data samples in order to remain on an IsoFLOP. The classical approach, based on the "Chinchilla" scaling laws (Hoffmann et al., 2022), involves training models *from scratch* for each compute budget and model size. The need to train models from scratch is due to the reliance on the cosine LR scheduler. Hoffmann et al. (2022) showed that the final performance of any pre-trained model is optimal when the cosine LR decay length matches the total training duration. This LR scheduler is also popular in other domains, including weather forecasting (Kurth et al., 2023; Bodnar et al., 2024). Several frontier NLP models as well as other SciML models have explored neural scaling laws in this manner (Grattafiori et al., 2024; DeepSeek-AI et al., 2024).

**Scaling with periodic cooldowns.** While the above approach is popular, it is expensive. It would be more beneficial to change the LR scheduler to a constant schedule, followed by a rapid cooldown phase to zero LR (Hägele et al., 2024). If one could show that the performance of this constant-plus-cooldown LR scheduler matches that of the cosine decay scheduler, then this would allow for training the series of models *just once*, considerably reducing the computational expense of neural scaling. We record the loss at a given compute budget by applying a LR cooldown to zero in the final pre-training iterations. To reach larger budgets, we simply resume from the checkpoint prior to the cooldown, continue training with the original LR, and then cooldown at the next target budget.

### 3.1 WEATHER FORECASTING PROBLEM FORMULATION

Our goal is to model (emulate) global atmospheric dynamics using a learned neural network by predicting the temporal evolution of the state $\mathbf{u}(\mathbf{x}, t)$, with $\mathbf{x} \in \mathbb{R}^2$ and $t \in [0, \infty)$. Our training dataset consists of discrete snapshots $\mathbf{u}_n$ sampled at regular time steps with interval $\Delta t$ with $\mathbf{u}_n \in \mathbb{R}^{H \times W \times C}$. Here, $H \times W$ is the height and width of the discretized latitude-longitude projection

$\mathbf{u}$; $C$ is the number of state variables (as wind velocities, temperature, and others; see §A.2, with vertical dimension treated as channels). The temporal evolution is modeled as:

$$\mathbf{u}_{n+1} = \mathcal{F}_\theta(\mathbf{u}_n), \tag{1}$$

where $\mathcal{F}_\theta$ is a neural network parameterized with weights $\theta$. We consider a standard mean-squared error (MSE) loss to pretrain $\mathcal{F}_\theta$:

$$\mathcal{L} = \min_\theta \sum_{B,H,W,C} (\mathcal{F}_\theta(\mathbf{u}_n) - \hat{\mathbf{u}}_{n+1})^2, \tag{2}$$

with $B$ representing the batch size of samples and $\hat{\mathbf{u}}$ is the target. Once trained, the model produces forecasts during inference for $N$ steps, $t_{n+i}$ with $i \in \{1, \cdots, N\}$, by autoregressively feeding each predicted state back as input to produce subsequent forecasts. Based on prior work, we consider two post-training strategies to better align the pre-trained model with the forecasting task:

1. *Adjusted MSE (AMSE)*: Pre-training datasets for weather contain time steps ($\Delta t$) that are undersampled; they are not fine enough to sufficiently resolve fine-scale atmospheric dynamics. With chaotic dynamics, this causes models trained with MSE to smooth high-frequency structure. To (partially) deal with this, we consider the adjusted MSE (AMSE) loss (Subich et al., 2025) (see §A.3), which disentangles decorrelation and spectral amplitude components of the error, enabling the model to retain high-resolution features.

2. *Autoregressive Rollouts (AR)*: Autoregressive inference introduces generalization error due to a distribution shift, as predicted states are used as inputs instead of samples from the training distribution (Brandstetter et al., 2023). To mitigate this, models are fine-tuned to autoregressively predict multiple time steps. While AR fine-tuning reduces forecast error over longer horizons, it often leads to overly smooth predictions due to the chaotic dynamics, causing models to behave like ensemble-mean forecasts (Price et al., 2023).

Existing approaches typically rely on heuristic schedules with multiple tunable components, including the number of autoregressive steps, learning rate schedules, and fine-tuning duration, for both AR and AMSE strategies. In this work, we re-purpose the cooldown period to implement either strategy, aligning the model to the downstream task: maintaining high spectral resolution via AMSE, or reducing long-horizon forecast errors via AR. This approach removes the need for manually designed heuristic schedules. This also eliminates the need to experiment with these during pre-training, as they can be applied efficiently after training.

## 3.2 ARCHITECTURE AND TRAINING

We employ a standard architecture based on the Shifted Window (Swin) Transformer (Liu et al., 2021). While many weather models introduce domain-specific modifications and customized components, we follow Willard et al. (2024), which demonstrates that standard transformer backbones can be highly competitive when trained well. Our input $\mathbf{u}_n$ is projected to the latent space $\mathbf{X} \in \mathbb{R}^{H/p \times W/p \times E}$ using a standard patch embedding layer with patch size $p$ and embedding dimensions $E$. We add a simple coordinate-based positional encoding using coordinates $(x, y, z)$ (derived from spherical coordinates) and normalized time step $t \in \mathbb{R}$ for each pixel, projected to the latent space $\mathbf{PE} \in \mathbb{R}^{H/p \times W/p \times E}$ via an additional patch embedding layer. The positional encoding is added to the latent feature $\mathbf{X}$ before being passed to the Swin Transformer blocks. While alternative positional encoding strategies exist, we avoid learnable encodings, which would introduce a large number of parameters at high input resolutions (especially for small patch sizes $p$) and risk overparameterizing this component relative to the rest of the model. We find this positional encoding to be sufficient in our experiments.

For the Swin blocks, we remove components such as relative positional bias and other non-essential modifications, including the hierarchical Swin structure (e.g., patch merging and downsampling), and we retain a uniform Transformer block design consisting of windowed multi-head self-attention (W-MHSA) and a multi-layer perceptron (MLP) with pre-RMSNorm and residual connections:

$$\tilde{\mathbf{X}} = \text{RMSNorm}(\mathbf{X}), \quad \mathbf{Y} = \text{W-MHSA}(\tilde{\mathbf{X}}) + \mathbf{X},$$
$$\tilde{\mathbf{Y}} = \text{RMSNorm}(\mathbf{Y}), \quad \mathbf{O} = \text{MLP}(\tilde{\mathbf{Y}}) + \mathbf{Y},$$

where the W-MHSA operation partitions the latent patch grid into 2D windows and computes standard dot-product self-attention locally. We use QK-normalization for training stability (Henry et al., 2020; Zhai et al., 2022). Every alternate block shifts the windows to enable cross-window interactions, implemented as a rolling operation with attention masks. We modify the mask to preserve the left-right periodicity of the Earth system. The final layer is a reverse patch embedding layer.

**Distributed training for neural scaling.** As model, dataset, and input sizes increase, standard data parallelism becomes insufficient, with per-GPU batch sizes dropping to one. While NLP models commonly address this using model parallelism, SciML domains such as weather forecasting face different challenges. High-resolution inputs and small patch sizes make these models heavily constrained by intermediate activation memory rather than weights (and optimizer), a limitation that is further amplified during autoregressive rollout (AR) in training. Standard techniques such as Fully-Sharded Data Parallelism (FSDP) do little to alleviate this memory pressure. Moreover, I/O pressure increases due to high-resolution data. It is essential to build efficient distributed infrastructure to train across a wide range of scales.

Towards this end, we implement spatial parallelism through domain decomposition, along with data parallelism. We employ a 2D array of $n_1 \times n_2$ GPUs orthogonal to the data parallel GPUs ($n_d$), to further partition the input ($H \times W$). Since activation memory scales as $O(BHWE)$, with batch size $B$ and embedding dimension $E$, this partitioning effectively reduces this memory to $O(BHWE/n_d n_1 n_2)$. The shifting window operation in the Swin require additional communication that we implement with a custom distributed rolling operation in the forward and backward pass (for details, see §A.3).

### 3.3 Empirical Analysis Setup

**Data.** Our models are trained on the ERA5 (Hersbach et al., 2020) dataset that consists of around $350,000$ samples of the Earth's atmospheric state from 1979 to 2022 at an hourly temporal resolution and a spatial resolution of $0.25°$. While ERA5 consists of hundreds of state (physical) variables, we use a 71 variable subset following Willard et al. (2024) (see §A.2). When projected on a latitude-longitude grid, each sample is a $71 \times 721 \times 1440$ tensor with $721 \times 1440$ representing the spatial grid. We use the years 1979 to 2016 as training data, 2017 as validation, and 2020 for testing. We train our models to predict a time step of 6 hours.

**Loss and Metrics.** Our pre-training loss is MSE. We avoid domain-specific weighting strategies that are variable dependent in the loss to maintain a simple objective for the neural scaling analysis. Our inputs are standardized mean and standard deviation derived per-variable from the training dataset. For validating our results, we use the area-weighted Root Mean-Squared Error (RMSE) and Power Spectral Density (PSD) prediction and target for each variable. We track each metric as a function of lead time up to 10 days, corresponding to the medium-range forecasting regime, and average the results over multiple initial conditions sampled across the evaluation period. We follow this evaluation protocol on initial conditions sampled regularly for the testing year 2020 (every 12 hours, 732 initial conditions) based on Rasp et al. (2024). We also retrieve the benchmark model predictions from Google Cloud Storage (using gcsfs), following Rasp et al. (2024). While RMSE provides a measure of average error over the globe, the PSD is commonly used to assess the effective resolution of the predictions and to quantify how well high-frequency, fine-scale features are captured. See §A.2 for the definitions.

**Training.** We train all our models with a batch size of 16. Through tuning experiments, we select a peak LR of 5E-4 for models larger than $100M$ parameters. Similar to NLP (Kaplan et al., 2020), we find smaller models can tolerate higher peak LRs, with the optimal LR scaling approximately with the square root of the parameter count. We train our models with the AdamW optimizer with a small weight decay of 1E-4 and gradient norms clipped at 1. We find that this, in conjunction with QK-normalization, is useful for training stability at the higher parameter counts. We use the constant+cooldown LR scheduler of the scaling experiments and cosine for initial comparisons. Both schedulers use 500 iterations as warmup. We use mixed precision with float16 for training. See Tab. A1 for all hyperparameter choices.

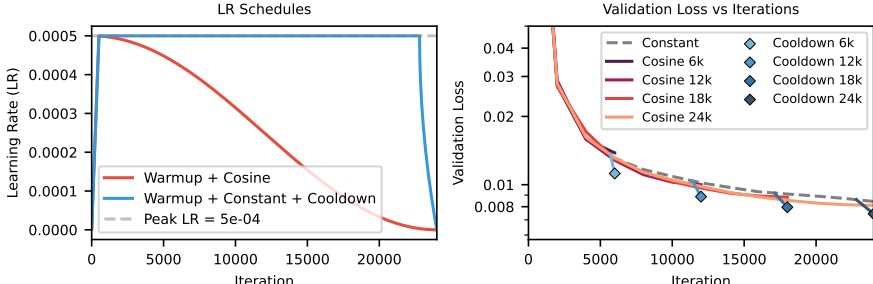

Figure 2: **Loss behavior for cosine vs constant LR with cooldown.** (left) LR schedules: The cosine schedule follows a half-cosine decay after a fixed warmup, while the constant+cooldown has a constant LR after the same fixed warmup, but then cools down rapidly to 0 at the end. The cooldowns happen at the last 5% of iterations. (right) Loss vs iterations for different `Swin` models: The validation loss of the model continuously trained with a constant LR and cooled down to different iteration counts (5% cooldown period) shows better losses compared to the `Swin` trained from scratch with different cosine schedulers that match the total iteration count.

## 4 MAIN RESULTS

We demonstrate the following results: **(R1)** Neural weather models can be continuously trained with periodic cooldowns using simple Transformer backbones; **(R2)** Cooldown periods can be repurposed with alternate losses to align the Transformer to a weather-specific downstream task; and **(R3)** Compute-optimal regimes, through systematic neural scaling using cooldowns, can be identified for weather forecasting with scaling laws across multiple compute budgets.

**(R1): Cosine vs. Cooldowns.** We show the two LR schedulers in Fig. 2. We first train a moderately sized $115M$ parameter `Swin` (see architecture hyperparameters in Tab. A1) from scratch multiple times using the cosine LR schedule, each run to a different number of iterations (batch size 16, corresponding to 20,819 iterations per epoch). We use the standard MSE loss with one-step prediction (predict the state $\Delta t = 6$ hours ahead). We then replace this setup with the constant+cooldown LR schedule, cooling down to the same final iteration counts as the cosine runs using only 5% of the total iterations. We observe that the models that have been cooled down consistently show lower validation loss compared to the cosine LR (see Fig. 2 for the validation loss curves for each run). Hence, this is a valid (and better performing) strategy for continual training, to carry out cheaper neural scaling experiments.

In order to use the cooldowns for neural scaling, we investigate what the optimal

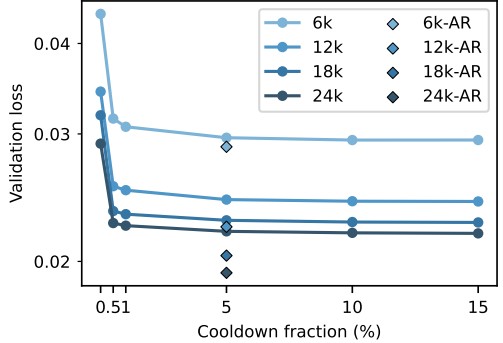

Figure 3: **Loss as a function of total iterations used for cooldown.** We show the MSE over 36 hours (6 autoregressive steps) of prediction averaged over the validation data (2017). The MSE loss decreases predictably with longer cooldown durations and this is true over multistep predictions. At around 5%, the gains start to diminish. The loss behavior also holds when the cooldown is repurposed with 4-step `AR` loss that allows for lower errors across the longer horizon.

fraction of the total iterations should be used for the cooldown period. We repeat the cooldown experiments using different cooldown fractions (from 0% to 15%) for each final iteration count. We note that while the one-step prediction is important for training, the true utility of a neural emulator lies in its ability to produce reliable forecasts over longer time horizons using autoregressive inference, as described in §3. Hence, we perform autoregressive rollouts during validation over multiple

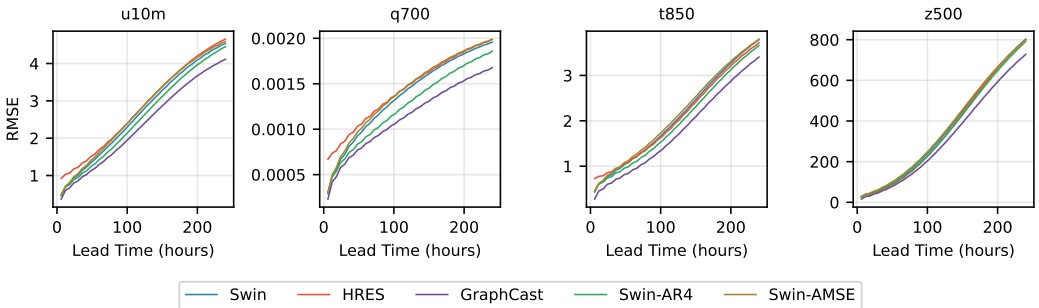

(a) RMSE of different variables (labeled at the top) vs. lead time (in hours) for different models.

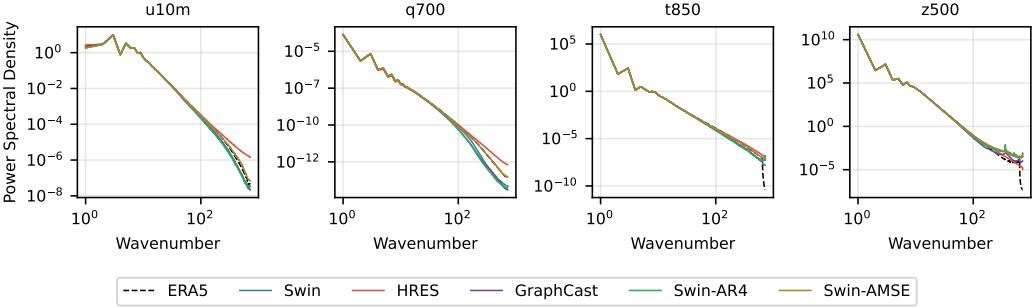

(b) PSD of different variables (labeled at the top) compared to PSD of the truth (ERA5) for different models at 24 hours lead time.

Figure 4: **Cooldowns can be used for alignment.** When evaluated on testing year 2020, the `Swin` cooled down at 24000 iterations is able to surpass the NWP (HRES) and is comparable to the state-of-the-art deterministic deep learning benchmark, Graphcast. When `AR` is used in cooldown (`Swin-AR4`), the RMSE drops further, consistent with the use of this loss. When `AMSE` is used (`Swin-AMSE`), the PSD retains high wavenumbers. This is easily seen in $q700$ where the `AMSE` spectra matches ERA5 perfectly, but other models blur significantly. `AR` contributes to more blurring in favor of reduced RMSE (visible in dissipation of power in high wavenumbers). We note that HRES models weather at a $0.1°$ resolution and hence shows higher resolution.

steps to track the time-averaged loss. Importantly, a cooldown that improves one-step loss but does not translate to longer-horizon accuracy may reduce its practical benefits. In Fig. 3, we track the validation loss over six steps (36 hours), and we find that the gains from cooldown translate to the downstream forecasting task as well. We observe that even a small cooldown fraction (0.5%) is able to quickly reduce the loss and also see that a cooldown of 5% is sufficient, beyond which the gains plateau. We use 5% cooldowns in all experiments henceforth.

**(R2): Alignment with Cooldown.** We demonstrate that the cooldown can be used to also align the model better with the downstream task of high-resolution forecasting for several time steps. This is important because neural scaling can be conducted using a simple `MSE` objective on large models and data, while the model's behavior can be adapted during the cooldown (or post-training) using alternative objectives. As a result, scaling experiments need not be repeated each time the loss function is modified to improve performance on specific scientific tasks. We consider both autoregressive training (`AR`) and the `AMSE` loss for this purpose (as described in §3). While `AR` aligns the model for long-term forecasting skill (by emulating an ensemble to produce smoother forecasts), `AMSE` aligns the models to produce high resolution features that are lost due to large timesteps.

In Fig. 3, we show that if the model is cooled down with 4-steps of `AR`, the validation loss systematically improves at all the iteration counts. In Fig. 4a, we show the RMSE of the `Swin` predictions as a function of lead time for several variables, compared to the NWP gold standard (HRES) (European Centre for Medium-Range Weather Forecasts, 2023) as well as the deterministic state-of-the-art emulator GraphCast (Lam et al., 2023). We see that the `Swin` is performant (outperforming HRES)

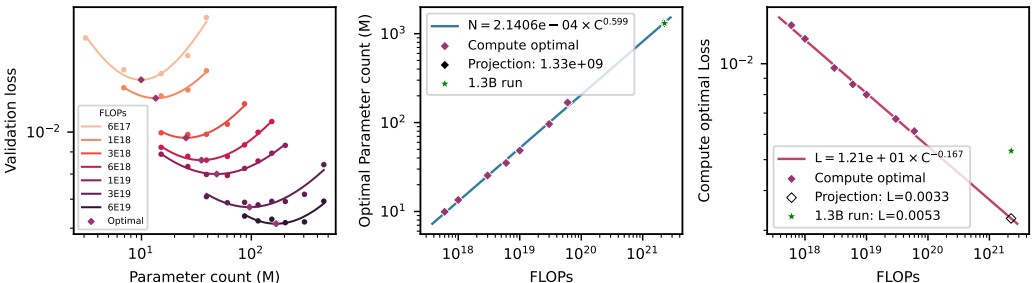

Figure 5: **Optimal model sizes as a function of compute.** (left) Similar to Fig. 1, we show the validation loss as a function of model size for different compute budgets. For each budget, the different model sizes are trained to a different number of iterations to create an IsoFLOP. We track the minima for each IsoFLOP (through a fitted parabola). (middle) We fit an empirical scaling law to find the optimal model size for any FLOP budget and project to 2.25E+21 FLOPs to find the optimal model. (right) We also project the loss to the final FLOP value—the measured loss at this FLOP value is saturated at 0.005.

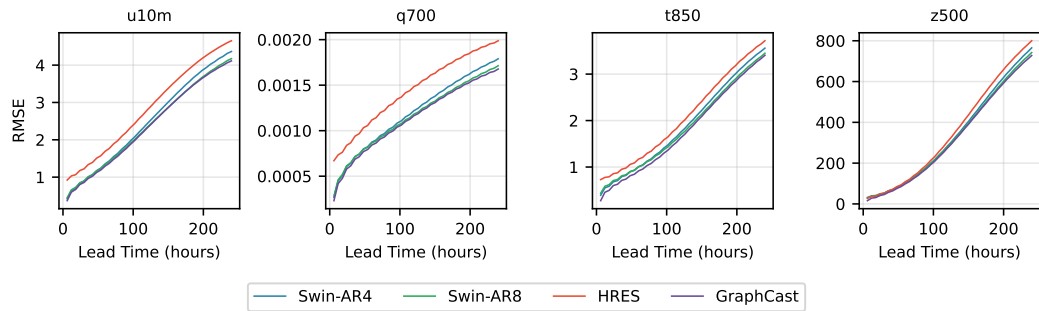

Figure 6: **RMSE performance of compute-optimal `Swin` vs baselines.** The 204M `Swin` (compute-optimal 6E+19 FLOPs) outperforms NWP / HRES and is on-par with GraphCast when evaluated over initial conditions in 2020. The model is cooled down with 4 step `AR`. With more steps (8 step `AR`), the performance gap reduces at longer lead times, as expected (GraphCast uses 2–12 step `AR`).

and can be systematically pushed to the GraphCast accuracy through `AR` (note that GraphCast was finetuned with 1 through 12-step `AR`). We can also see from the PSD metric in Fig. 4b that the RMSE gains occur through smoothing that emulates an ensemble prediction. Similarly, using `AMSE` during cooldown, we see that RMSE remains largely unaffected; however, the PSD plots show that the model has more power in the higher wavenumbers that this loss was designed to capture. We emphasize that the goal here is not to achieve best possible performance, but rather to demonstrate that post-training phases such as cooldowns can be re-purposed to align model behavior in weather forecasting, highlighting their flexibility without entangling them with the core scaling analysis.

**(R3): Neural Scaling.** We train a range of model sizes from $3M$ to $456M$ parameters (see all hyperparameters in Tab. A1), varying more than two orders of magnitude. We select two orders of magnitude of compute budgets from 6E+17 to 6E+19. For each budget, we estimate (see §A.3 for details on FLOP estimates) a range of model sizes and number of iterations to train such that each configuration lies on an IsoFLOP. Each model is trained only once and cooled down to the required iteration at each IsoFLOP. We note that several of our models train for more than one epoch. In Fig. 1, we show the IsoFLOP curves on the loss versus number of samples scaling plot, while Fig. 5 presents the corresponding scaling behavior of loss versus model size. For both plots, we fit parabolas to each IsoFLOP to estimate the compute-optimal dataset size and model capacity for a given budget. We observe that *(i)* the model scales to 6E+19 plots without saturating the loss and *(ii)* exhibits compute-optimal behavior where each budget affords an optimal model size and sample count. Interestingly, this trend persists even at 6E+19 where the model is trained for multiple epochs. In Fig. 6, we show that the model that is closest to compute-optimal at 6E+19

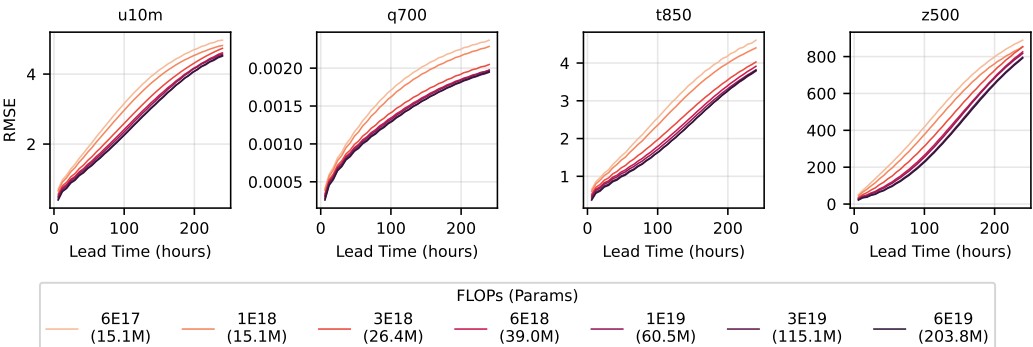

Figure 7: **RMSE performance of the closest compute-optimal `Swin` models at each compute budget.** RMSE at extended forecast horizons, up to 240 hours, consistently improves with increased compute. At the highest compute levels, the rollout performance begins to exhibit signs of saturation.

FLOPs ($204M$) matches the state-of-the-art GraphCast. We find that the accuracy metrics maintain their gains throughout the rollout timesteps up to 240 hours and across different compute budgets. In Fig. 7, we present the RMSE over the time horizon using the closest compute-optimal model at each budget. As the compute increases, accuracy improves for both one-step predictions (as shown in Fig. 5) and over longer time horizons. At the highest budgets, however, we start to see a saturation.

We then derive the following scaling: $S^\star(C) \propto C^{0.41}$ and $N^\star(C) \propto C^{0.59}$, where $C$ is the budget, $S^\star$ is the optimal number of (pseudo) samples (with the understanding that multiple epochs are possible) and $N^\star$ is the optimal model size (see Fig. 5). We extrapolate to 2.25E+21 FLOPs (more than an order of magnitude from the scaling analysis) to investigate how long the scaling holds. The empirical scaling suggests training a $1.3B$ parameter model for $272K$ iterations and we train this large model. We observe in Fig. 5 that this model (denoted with star) saturates before hitting the projected loss value. We attribute this saturation to overfitting as this model needs to be trained to more than 13 epochs to reach this compute budget. We show this in Fig. A8 by comparing the training and validation loss curves for all models. This provides early evidence that scaling trends may begin to break down in this regime, strongly motivating this kind of neural scaling analysis before moving SciML models to frontier model scales, data scales, and resolutions.

## 5 CONCLUSIONS

We systematically studied neural scaling for data-driven weather forecasting SciML models. To do so, we used a minimalist Transformer backbone, without any domain-specific complexities, so that scaling behavior could be studied in its simplest and most interpretable form, without confounding architecture / loss function choices. We used a continual training approach with constant LRs and periodic cooldowns, an approach which offers several advantages. This outperforms the commonly-used cosine decay schedule; it enables rapid alignment of the model with downstream objectives during short cooldown phases; and, most importantly, it makes scaling law analysis practical by allowing IsoFLOP curves to be constructed without repeatedly training models from scratch. We demonstrated this by efficiently exploring a wide range of model sizes and compute budgets, identifying compute-optimal relationships between model size, dataset size, and training compute, with neural scaling trends emerging across two orders of magnitude in compute (even with multi-epoch training). Our results indicate that extrapolating to large compute budgets with models exceeding a billion parameters shows signs of saturation. This is due to the limited dataset size and the need to train for multiple epochs to reach these compute budgets. Even when following compute-optimal scaling, large models require multiple passes over the data, so simply increasing model size may not yield proportional gains. These observations suggest that very large SciML models should be carefully analyzed and planned before scaling further. Overall, our results emphasize that understanding neural scaling behavior for a given dataset is critical before training SciML models at frontier scale, and they provide a practical framework for doing so in spatiotemporal scientific domains.

ACKNOWLEDGEMENTS

This research used the Perlmutter supercomputing resources of the National Energy Research Scientific Computing Center (NERSC), a U.S. Department of Energy Office of Science User Facility located at Lawrence Berkeley National Laboratory, operated under Contract No. DE-AC02-05CH11231. An award of computer time was provided by the ASCR Leadership Computing Challenge (ALCC) program at NERSC under ERCAP0038267. SS would like to thank Ankur Mahesh for helpful discussions on weather modeling and downstream metrics. Our conclusions do not necessarily reflect the position or the policy of our sponsors, and no official endorsement should be inferred.

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

# A    APPENDIX : ADDITIONAL DETAILS

## A.1    LIMITATIONS

The main limitation of this paper is its focus on deterministic training of the weather model. Given the inherently chaotic nature of weather, probabilistic estimates are often more informative for forecasting at future time points, with metrics such as continuous ranked probability scores (CRPS) and spread-skill ratios (SSR) providing better diagnostics for a model's predictive capability. Addressing this would involve training (or cooling down) models using diffusion objectives (Price et al., 2023; Alexe et al., 2024) or CRPS-based ensemble training (Alexe et al., 2024; Bonev et al., 2025), and this may exhibit different scaling behaviors—this is left for future work. Our patch size is fixed for the scaling tests, but the internal Transformer resolution (sequence length) represents another dimension for systematic scaling studies. Another possible limitation is the focus on Transformer architectures. While this focus is intentional to demonstrate that simple architectures can scale and perform competitively, it remains important to explore other architecture types, such as FourCast-Net Bonev et al. (2025), which incorporates geometric inductive biases through spherical transforms within the neural operator framework, or graph-based approaches like GraphCast Lam et al. (2023). Finally, to rigorously probe the implications of scaling laws and not be constrained by multi-epoch training, it is essential to extend beyond this single domain, incorporating diverse datasets from across the Earth Sciences (and beyond), and exploring cross-domain foundation model pre-training. This will allow us to evaluate scaling behavior in a broader, more complex data landscape.

## A.2    ERA5 DATASET AND METRICS

For the current weather state $u_n \in \mathbb{R}^{71 \times 721 \times 1440}$, we closely follow existing works (Bonev et al., 2025; Willard et al., 2024) and select the following variables: geopotential height ($z$), winds ($u$, $v$), temperature ($t$), and specific humidity ($q$) at 13 vertical pressure levels (50hPa, 100hPa, 150hPa, 200hPa, 250hPa, 300hPa, 400hPa, 500hPa, 600hPa, 700hPa, 850hPa, 925hPa, and 1000hPa). We also include the following surface variables: $u10m$ ($u$ at 10m), $v10m$ ($v$ at 10m), $t2m$ (temperature at 2m), surface pressure ($sp$), mean sea level pressure ($msl$), and total column water vapor ($tcwv$). Finally, we include land-sea mask, orography, and cosine of the latitude (to convey the curvature of the earth) as static (not dependent on time) masks (not predicted).

Our metrics are area-weighted RMSE and PSD for each channel $C$ defined as follows. Given prediction $u_n$ and ground truth $\hat{u}_n$ at time step $n$,

$$\mathrm{RMSE}(u_n, \hat{u}_n) = \sqrt{\frac{1}{HW} \sum_{h=1}^{H} \sum_{w=1}^{W} w_h \big(u_{c,h,w} - \hat{u}_{c,h,w}\big)^2},$$

with $w_h$ being the area weights, proportional to the cosine of the latitude, normalized to one.

The PSD (Wik; Bonev et al., 2025) of $u$ at spherical harmonic degree $\ell$ is defined as:

$$\mathrm{PSD}_\ell(u) = \frac{1}{2\ell + 1} \sum_{m=-\ell}^{\ell} |u_{\ell m}|^2, \tag{3}$$

where $u_{\ell m}$ are the coefficients of $u$ in the spherical harmonic decomposition:

$$u(\theta, \phi) = \sum_{\ell=0}^{\infty} \sum_{m=-\ell}^{\ell} u_{\ell m} Y_{\ell m}(\theta, \phi),$$

with $Y_{\ell m}$ denoting the spherical harmonic functions of degree $\ell$ and order $m$, and $(\theta, \phi)$ the latitude and longitude coordinates. We use `torch-harmonics` (Bonev et al., 2023) to implement the Spherical Harmonic Transform (SHT). We apply all SHTs in FP64 to avoid numerical issues. We use the SHTs for the metrics as well as the `AMSE` loss functions, defined below.

For the `AMSE` loss (Subich et al., 2025) between $u$ and $\hat{u}$, we have:

Table A1: Model configurations used for the neural scaling: We run each of these models once to different final iteration counts based on the required FLOP budgets with 5% cooldown periods at the end for the LR.

| Param (M) | Embed | Depth | LR |
|-----------|-------|-------|--------|
| 3.1 | 192 | 6 | 1.5e-3 |
| 6.9 | 256 | 8 | 1.5e-3 |
| 15.1 | 384 | 8 | 1.5e-3 |
| 26.4 | 512 | 8 | 1.2e-3 |
| 39.0 | 512 | 12 | 1e-3 |
| 60.5 | 640 | 12 | 8e-4 |
| 86.8 | 768 | 12 | 6e-4 |
| 115.1 | 768 | 16 | 5e-4 |
| 153.5 | 1024 | 12 | 5e-4 |
| 203.8 | 1024 | 16 | 5e-4 |
| 304.5 | 1024 | 24 | 5e-4 |
| 456.7 | 1536 | 16 | 5e-4 |

$$\text{AMSE}(u, \hat{u}) = \sum_{k} \left( \left( \sqrt{\text{PSD}_k(u)} - \sqrt{\text{PSD}_k(\hat{u})} \right)^2 + 2 \max \left( \text{PSD}_k(u), \text{PSD}_k(\hat{u}) \right) \left( 1 - \text{Coh}_k(u, \hat{u}) \right) \right),$$

$$(4)$$

with the coherence between $u$ and $\hat{u}$ at degree $\ell$ as:

$$\text{Coh}_\ell(u, \hat{u}) = \frac{\sum_{m=-\ell}^{\ell} \text{Re}\left( u_{\ell m} \, \hat{u}_{\ell m}^* \right)}{\sqrt{\text{PSD}_\ell(u) \, \text{PSD}_\ell(\hat{u})}}.$$

$$(5)$$

### A.3 MODEL HYPERPARAMETERS AND TRAINING DETAILS

We train the `Swin` models (each with global batch size 16) summarized in Tab. A1. We find that the small models can handle larger LRs and we increase LRs for the smaller models up to $1.5E - 3$. For the larger models, we decrease until $5E - 4$. The LR (inversely) scales approximately with the square root of the parameter count in our tuning experiments. We fix our patch size to 4 based on previous studies patch size experiments (Willard et al., 2024). However, we acknowledge that this is an additional moving part that we have not considered in this study. We also fix the head dimension to 64 through minimal hyperparameter tuning experiments as well as the window size to $9 \times 18$ based on Willard et al. (2024). We observe minimal accuracy gains with larger windows and smaller ones begin to degrade the accuracy. For the feedforward (MLP) layers, we use a hidden size to embedding dimension ratio of 4 for all models. For the $1.3B$ `Swin`, we use an embedding dimension of 1792 and depth 34. For the cooldowns, we use a 1-sqrt() cooldown function based on Hägele et al. (2024). Given the high resolution ($64800 = 180 \times 360$ sequence length from patch 4), we can only train most models with a local batch size of one. All our models are trained on NVIDIA A100 GPUs (40G memory capacity, unless specified). For models up to $100M$, we use only data parallelism (16 GPUs). For larger models, we employ various degrees of spatial parallelism using an orthogonal array of GPUs. For the largest model ($456.7M$ parameters) in neural scaling, we use $2 \times 2$ spatial parallel (4 GPUs) for a total of 64 GPUs. For AR cool downs with 4 steps, the memory complexity increases linearly and hence this model would then require 16 GPUs for spatial parallel and a total of 256 GPUs. For only the $1.3B$ parameter model, we use 80G A100s and train it on 64 GPUs with spatial parallel 4.

**Hybrid spatial (domain) + data parallelism.** The main ingredient in spatial parallelism is implementing a distributed roll operation. Instead of shifting the windows directly, the original `Swin` implementation (Liu et al., 2021) simply rolls (cylic-shifts) the image instead and applies an attention mask to limit the attention to each sub-window. When the image is partitioned across GPUs, this roll operation must happen across the GPUs that shard the image. Hence, these spatial parallel GPUs

---

**Algorithm 1** Distributed Roll for Shifting Windows in `Swin`

---

**Require:** Roll dimension $d$, local tensor $x$ (spatially sharded in $d$ across GPUs), shift size $s$, spatial parallel GPU group $\mathcal{G}$
**Ensure:** Tensor $x$ rolled along dimension $d$
 1: $(r, P) \leftarrow \text{rank\_and\_size}(\mathcal{G})$ ▷ GPU rank and world size
 2: **if** $P = 1$ **or** $s = 0$ **then**
 3:    **return** $\text{roll}(x, s, d)$ ▷ purely local case
 4: $\text{right} \leftarrow (r + 1) \bmod P$
 5: $\text{left} \leftarrow (r - 1) \bmod P$ ▷ neighboring GPUs in the ring
 6: **if** $s > 0$ **then**
 7:    $\text{send} \leftarrow x[\ldots, -s:, \ldots]$ ▷ right boundary slice
 8: **else**
 9:    $\text{send} \leftarrow x[\ldots, :s, \ldots]$ ▷ left boundary slice
10: $\text{recv} \leftarrow \text{zeros\_like}(\text{send})$ ▷ receive buffer
11: $\text{distributed\_all\_to\_all\_single}(\text{recv}, \text{send}, \mathcal{G})$ ▷ $s > 0$ sends right / receives left
                                                              ▷ $s < 0$ vice versa
12: $x \leftarrow \text{roll}(x, s, d)$ ▷ local roll (boundary will be corrected)
13: **if** $s > 0$ **then**
14:    $x \leftarrow \text{concat}(\text{recv}, x[\ldots, -s:, \ldots], d)$
15: **else**
16:    $x \leftarrow \text{concat}(x[\ldots, :s, \ldots], \text{recv}, d)$
17: **return** $x$

---

must send and receive data from neighbors for an accurate roll. This boils down to a *collective permute* operation and we implement it using PyTorch's `torch.distributed.all_to_all_single` collective. Following frontier NLP research codes from `Megatron-LM` (Shoeybi et al., 2019), we implement the forward and backward pass of this distributed operation using PyTorch's custom `Autograd` functions, where the backward pass is simply the conjugate forward operation: a reverse distributed roll. We show the distributed roll algorithm in Alg. 1. We note that this is not a full replacement for distributed `torch.roll`, as we assume we only need to send one slice to one neighboring GPU and receive one slice from another neighboring GPU. This is true for `Swin`, because we shift by a half of the window size, and windows are not sharded across GPUs in our setting (there is at least one window on every GPU). For brevity, we also omit the `permute` operations to bring the shift dimension to position 0 and back (`all_to_all_single` only acts on the first dimension).

Finally, spatial parallel introduces weights (and bias) tensors that are *shared* across the spatial parallel GPU groups (in addition to the data parallel GPUs that, by design, share the model weights). Hence, the weight gradients (`wgrads`) must be additionally reduced in this group (along with the data parallel group). Our simplistic `Swin` model design (with coordinate position embeddings) leads to all the weights (and bias) tensors shared across the spatial parallel group and hence we implement this as a joint `AllReduce` operation across the combined data and spatial parallel GPU group using a communication hook in PyTorch's `DistributedDataParallel`. We verify with Unit Tests that the distributed forward and backward passes (including weight gradients and adjoints) match the non-distributed version ensuring accuracy of our custom model parallelism. Finally, we also use `NVIDIA TransformerEngine` (NVIDIA) for easy access to fused operations and support for 4D parallelism with tensor parallelism. While we do not use tensor parallelism in any run, we note that the code supports this parallelism as well for scaling up. Here, an additional orthogonal array of GPUs may be employed to further shard the model weights (similar to NLP). `TransformerEngine` provides a drop-in replacement for `Linear` layers in our model to use tensor parallelism. We note that some `bias` tensors may be shared across tensor and spatial parallel GPU groups with this additional parallelism and we take care of this through custom communication hooks in `DistributedDataParallel`. Unit tests confirm that combined spatial, tensor, and data parallelism yeild the correct distributed forward pass outputs and backward pass `wgrads` and `dgrads`.

**Efficient and stateful dataloaders.** Our data is stored in HDF5 files, with one file per year of ERA5, to efficiently leverage the Lustre filesystem in our experiments for high-throughput data access. We use `NVIDIA DALI` (NVIDIA, 2026) dataloaders, which automatically overlap data loading

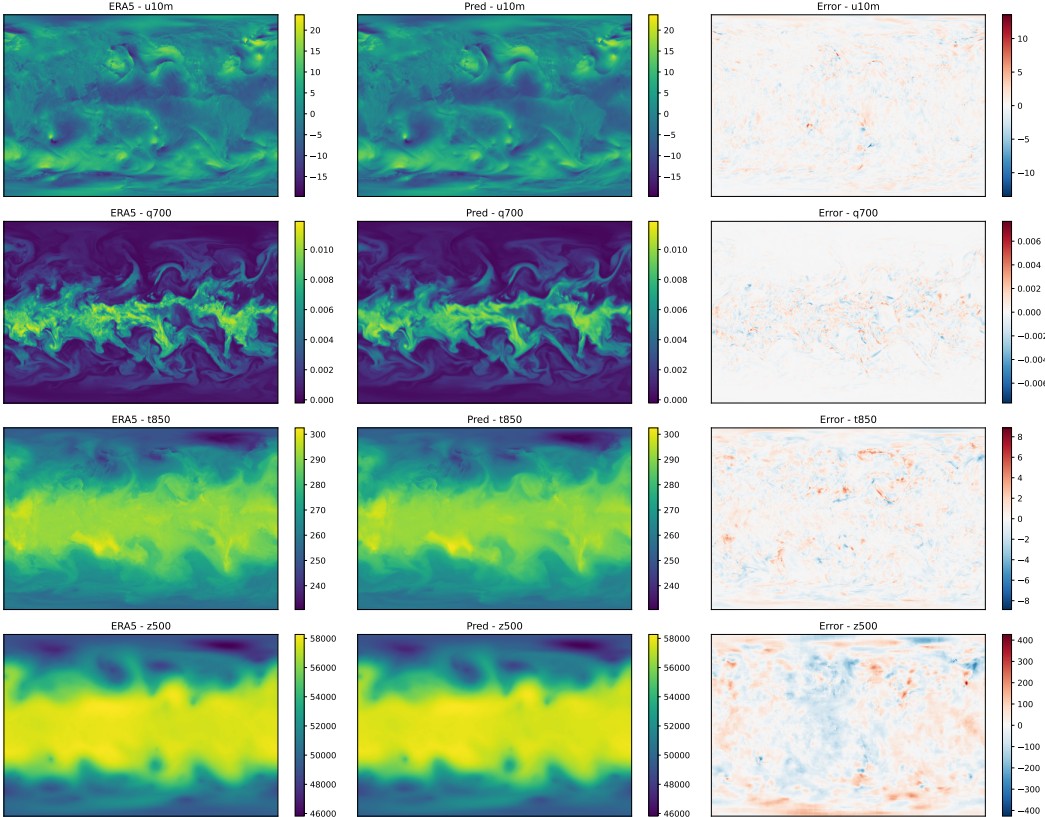

Figure A1: **Predictions from the 204M parameter model:** We show the ground truth ERA5, the prediction, and the error in physical units for four variables. The predicted forecasts also qualitatively match the ground truth.

with computation during the forward and backward passes and perform pre-processing operations, such as data normalization, directly on the GPU. This ensures that data I/O does not become a performance bottleneck. Finally, we train our models to fixed iteration counts (to support periodic cooldowns under a predefined FLOP budget), which requires checkpoint–restart functionality to operate correctly mid-epoch. To enable this, we implement stateful dataloaders within DALI that track both the shuffled sample indices and the number of completed iterations, allowing training to resume seamlessly from arbitrary mid-epoch checkpoints.

**Code.** We open-source our code at `https://github.com/ShashankSubramanian/neural-scaling-weather`. For all our FLOP computations, due to the homogeneous design of the `Swin`, we use standard analytical attention and MLP FLOP estimates, taking into account the windowed attention, and verify with `fvcore` (Facebook AI Research, 2023) that our analytical values are close to the correct estimates. We include the patch, position, and reverse-patch embedding FLOPs as well in our estimation. The FLOP expressions can be found in `utils/flops_utils.py` in our code.

## A.4 Additional results

We visualize the prediction from the compute-optimal model ($203.8M$) at the large budget 6E+19 in Fig. A1, along with the bias (error) between the predictions and the ground truth (ERA5). The predictions are obtained on the testing year 2020, with initial conditions every 12 hours (for a total of 732 initial conditions). Beyond the quantitative metrics defined earlier, we observe that the model produces realistic predictions of the physical fields, capturing the main spatial structures and patterns present in the reference data.

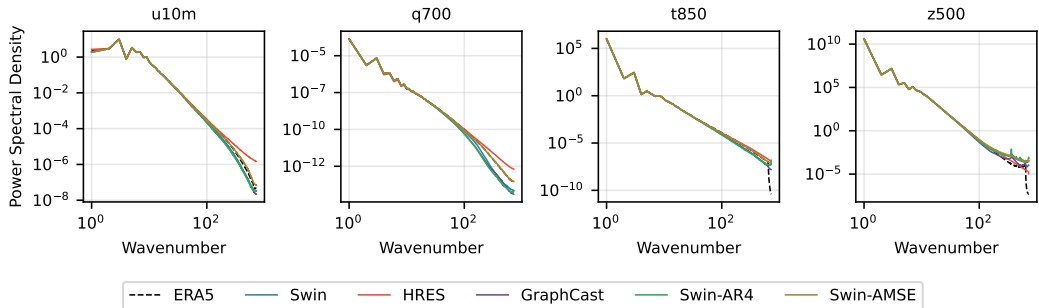

(a) PSD of different variables compared to PSD of the truth (ERA5) for different models at 24 hours lead time.

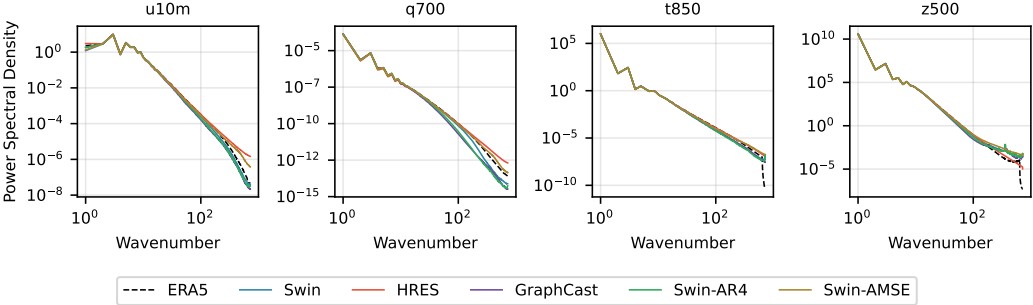

(b) PSD of different variables compared to PSD of the truth (ERA5) for different models at 120 hours lead time.

Figure A2: **PSD of predictions at 1 day and 5 days:** While `AR` produces blurrier forecasts (though with lower RMSE), `AMSE` helps retain power at high wavenumbers. At longer lead times, such as 120 hours, `Swin-AMSE` continues to show significant power at these small scales. This allows practitioners to define an effective resolution based on this noise-level as outlined in Subich et al. (2025).

All the `Swin` models fit on a single GPU (A100 40G) during inference. Most models generate 10 day forecasts in less than a minute (including metric computations). For example, the $204M$ `Swin` takes 32 seconds, while the largest $1.3B$ `Swin` takes around 96 seconds to generate the same 10 day forecast. These models preserve the emulator's substantial computational advantage over the NWP, achieving a speedup of more than $100\times$ (Alexe et al., 2024).

In Fig. A2, we show the PSD of different variables for two lead times (1 day and 5 days). The `Swin` model cooled down with `AR4` loses a significant amount of energy at small scales, resulting in overly dissipated forecasts and an effective resolution much coarser than what the $0.25°$ grid can support. In contrast, the `Swin` model cooled down with `AMSE` matches the PSD of the ground truth almost exactly at short lead times. At longer lead times, it exhibits higher energy than the ground truth at the smallest scales suggesting a noise-driven effective resolution that exceeds that of the dissipated `AR4` forecasts, as noted in Subich et al. (2025). Overall, the two cooling strategies guide the model toward different objectives: one prioritizes lower overall RMSE (and skill) over long lead times, akin to ensemble forecasting, while the other emphasizes high realism, preserving small-scale features in the forecast. In Fig. A3 and Fig. A4, we show the PSD for the closest compute-optimal model at each compute budget for a 24-hour and 120-hour lead time. While all models faithfully capture the PSD at the lower wavenumbers, the higher compute models show better PSD at the higher wavenumbers as well, better capturing the sharper features. However, improvements with increasing compute are gradual and show signs of saturation. Additionally, lower-compute models exhibit artificial power at high wavenumbers, which becomes more pronounced at longer lead times (120 hours). In contrast, higher-compute models maintain more consistent performance across lead times.

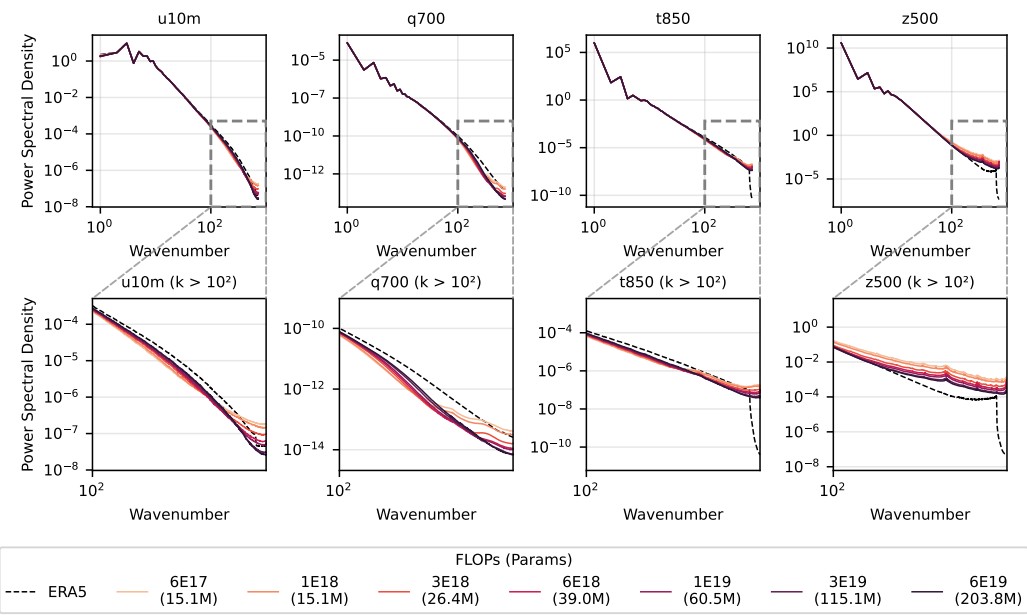

Figure A3: **PSD performance at lead time 24 hours of the closest compute-optimal** `Swin` **models at each compute budget:** As in Fig. 7 that shows consistent improvement of rollout RMSE with increased compute, the PSD also exhibits improvements, though they are smaller in magnitude and progress more gradually. Here, the PSD is at 24 hour lead time (averaged over the test year). In the bottom row, we zoom into the high wavenumbers and can observe that the bigger (more compute) models show sharper features and resemble ERA5 closer.

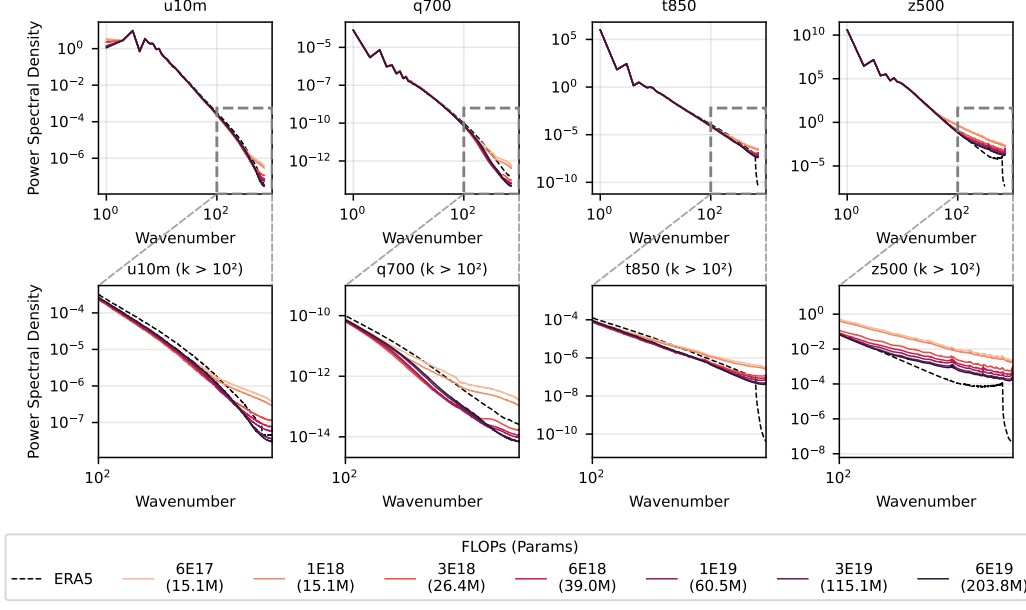

Figure A4: **PSD performance at lead time 120 hours of the closest compute-optimal** `Swin` **models at each compute budget:** The PSD is evaluated at a 120-hour lead time and averaged over the test year. All models accurately capture the low wavenumbers, while higher wavenumbers show gradual improvement as compute increases. The bottom row zooms in on the high-wavenumber range, where larger models exhibit spectra that more closely match ERA5, whereas smaller models appear more blurred and display some artificial high-wavenumber features. These features are more dominant at this longer lead time compared to 24 hours in Fig. A3.

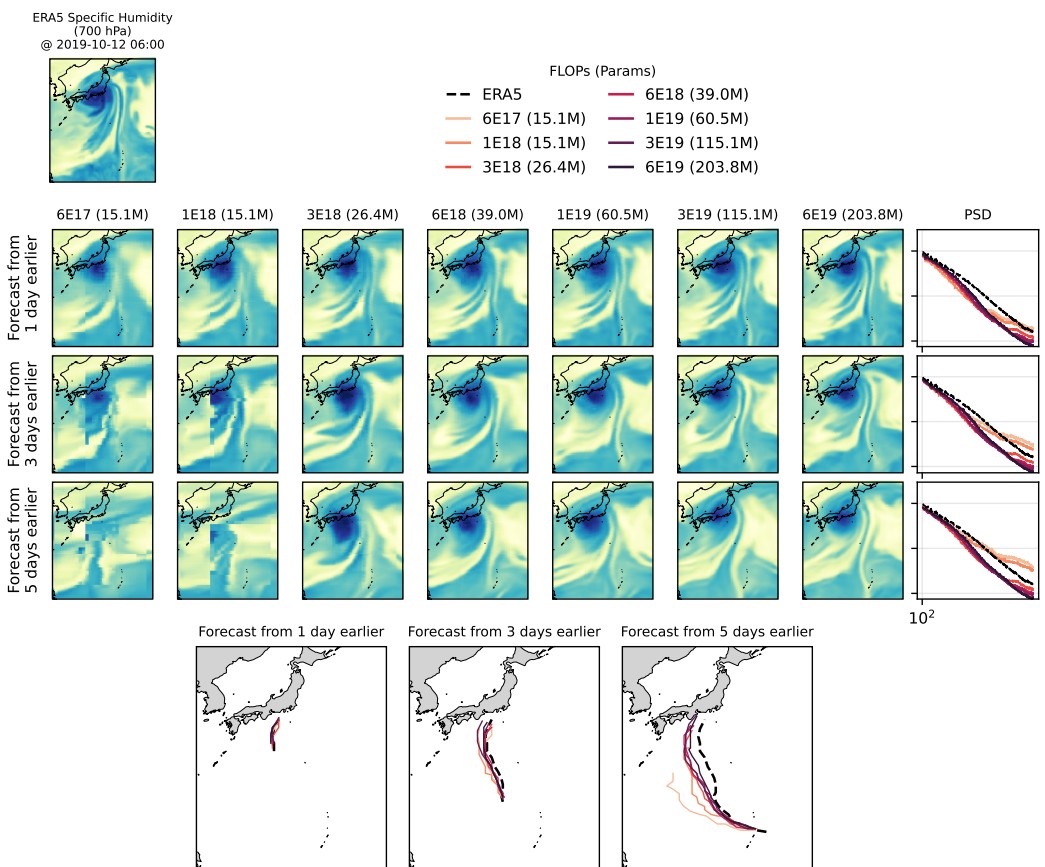

Figure A5: **Tropical Cyclone (TC) visualization and tracks before Typhoon Hagibis made landfall in Japan**: We reproduce the TC analysis from Price et al. (2023). We show the ERA5 ground truth for specific humidity at 700hPa on October 12, 2019, 06 UTC in the top left. The following three rows show the `Swin` predictions at the same time from all the closest compute-optimal models (between 6E17 and 6E19 FLOPs, initialized 1, 3, and 5 days earlier, respectively. The last column shows the PSD for the predictions (zoomed into the high wavenumbers). Forecasts initialized 1 day earlier show that all models capture the TC structure, although the larger models produce systematically sharper features, which is also reflected in the PSD. For initializations 3 and 5 days earlier, the smaller models fail to capture the TC structure, exhibiting patching artifacts and generally inconsistent blending of physical features. This behavior is likewise evident in the PSD as artificial high-wavenumber energy. In contrast, the larger models consistently capture the TC across all lead times. The last row shows the TC track from the different initializations. Although a deterministic model is insufficient for accurately assessing track performance, some variability among the different models is seen.

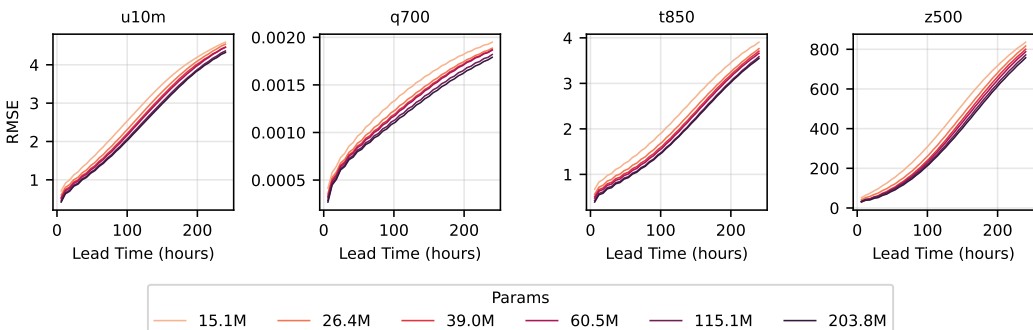

Figure A6: **Cooldown with `AR`4 is valid for different model sizes.** `AR`4 gains consistently improve with model size (and compute), similar to Fig. 7 that shows the same improvements in the `AR`1.

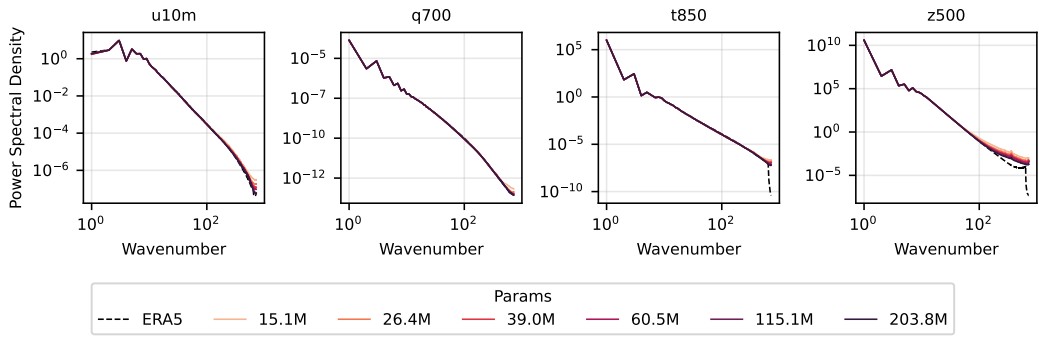

Figure A7: **Cooldown with `AMSE` is valid for different model sizes.** `AMSE` allows all models to match the ERA5 PSD more closely. PSD is computed at 24h lead time.

These spectral improvements are more visible in a specific physical case study. See Fig. A5. We reproduce the study that analyzes the Tropical Cyclone (TC) patterns of Typhoon Hagibis from (Price et al., 2023). We use the same TC tools as the study—TempestExtremes Ullrich et al. (2021), in order to visualize TC tracks. Here, we track the TC structure across the compute-optimal models initialized at 1, 3, and 5 days from the TC landfall time on October 12, 2019, 06 UTC. We visualize the specific humidity at 700hPa for all the forecasts as well as the PSD (compared to ERA5). For short forecasts (1 day), all models are faithful with increased compute resulting in sharper predictions (also quantified in the PSD). With longer forecasts (3 and 5 days earlier), the smaller models exhibit artificial features (patch artifacts, discontinuities in the prediction) that is also seen in the spectral power at the high wavenumbers. With more compute, the models begin to produce realistic TCs at these lead times as well. We also visualize the TC tracks. While all models follow the general track of ERA5, the smaller models show more variability at longer lead times due to artifacts developing. We note that, due to the deterministic modeling framework, these results should be interpreted as illustrative. Accurately tracking such phenomena, particularly at longer lead times, requires probabilistic (ensemble) forecasting, as the chaotic nature of atmospheric processes naturally introduces more spread (variance) in the predictions, particularly at longer lead times.

With respect to the cooldown methodology, Fig. A6 and Fig. A7 show that repurposing cooldowns with either `AR` or `AMSE` is valid across different model sizes, as well. This demonstrates the generality of the approach, allowing models to be aligned with different downstream objectives during the cooldown period.

Finally, in our extrapolated scaling to $1.3B$ parameters, we observe a saturation of performance as seen in Fig. 5. This is attributed to around 13 epochs of training and hence high chances of overfitting with the large (high parameter count) model. We show this in Fig. A8 where for all the models in the scaling regime (6E+17 to 6E+19) the training and validation loss track closely. However, for the model extrapolated to 2.25E+21 FLOPs, while the training loss reaches close to the extrapolated

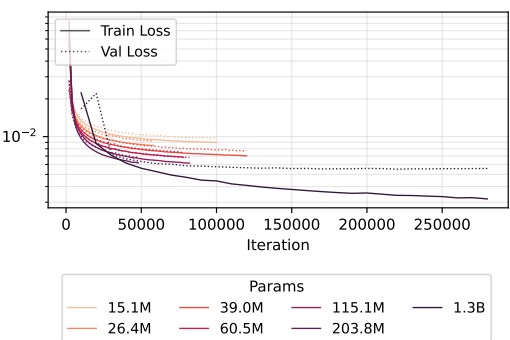

Figure A8: **Training and validation loss as a function of iterations for different models:** We show the training curves for all the (closest) compute-optimal models. All losses were logged at a frequency of 2000 iterations except the 1.3B model that was logged at a frequency of 10,000 iterations. The training and validation loss curves track each other closely. However, for the 1.3B model, we observe significant overfitting.

loss value of 0.033, the validation loss plateaus at around 0.053, pointing to significant overfitting. Although the validation loss continues to improve slightly with more compute, the growing gap suggests increased memorization of training data, which may reduce generalization abilities as well.

