# OpenReview forum: "On Neural Scaling Laws for Weather Emulation through Continual Training"
_ICLR.cc/2026/Workshop/FM4Science — ICLR 2026 Workshop FM4Science Poster_

### Official Review · Reviewer_MNii · 2026-02-13
**Systematic Characterization of Neural Scaling Laws for Weather Emulation via Continual Learning**

**Rating:** 8
**Confidence:** 4

**Review:**

**Summary**

This paper investigates neural scaling laws within the context of data-driven weather forecasting. While scaling laws are well-established in NLP and CV, the authors address a gap in scientific machine learning (SciML) where joint relationships between model size, data volume, and compute budget remain under-explored.

The authors propose a continual learning training strategy using constant learning rates and periodic cooldown phases. This approach allows for the efficient construction of IsoFLOP curves without the prohibitive cost of training every configuration from scratch. Using a standard Swin Transformer backbone, the study identifies compute-optimal training regimes across two orders of magnitude of compute ($6 \times 10^{17}$ to $6 \times 10^{19}$ FLOPs). They further demonstrate that cooldown periods can be "re-purposed" to align models with downstream tasks, such as improving multi-step rollout accuracy or capturing high-resolution features through spectral loss adjustments.

**Pros**

*1. Methodological Efficiency:* The use of continual learning with periodic cooldowns significantly lowers the barrier for conducting scaling experiments in SciML.

*2. Practical Downstream Alignment:* Re-purposing the cooldown phase for Autoregressive (AR) rollouts or Adjusted MSE (AMSE) effectively disentangles core scaling from task-specific fine-tuning.

*3. Infrastructure Contributions:* The implementation of 2D spatial parallelism through domain decomposition is a critical technical contribution for handling the high-resolution activation memory constraints unique to weather data.

*4. Empirical Rigor:* The authors provide a clear compute-optimal relationship ($N^*(C) \propto C^{0.59}$ and $S^*(C) \propto C^{0.41}$) and test the limits of these laws by training a 1.3B parameter model, identifying potential saturation points.

**Cons**

*1. Thematic Isolation from Related DL Dynamics:* The paper currently lacks a discussion on how these empirical scaling laws interact with established theoretical frameworks for model selection and fine-tuning. For instance, the study would benefit from referencing work such as LENSLLM (Zeng et al., 2025), which explores Hessian-based PAC-Bayes generalization bounds and NTK-based models to understand fine-tuning dynamics—a necessary theoretical complement to the empirical observations made here.

*2. Resolution and Data Saturation:* The observed saturation in the 1.3B model trained at $0.25^\circ$ resolution suggests that current scaling laws may be bottlenecked by dataset resolution. The authors should more explicitly compare these limits to other frontier models like GraphCast (Pritzel et al., 2023).

Overall, the work is of high quality and presents a systematic, reproducible framework for SciML scaling. The clarity of the writing and the transparency of the FLOP calculations make the findings actionable for other researchers. Its originality lies in adapting "Chinchilla-style" scaling to the spatiotemporal domain and introducing the flexible cooldown re-purposing. This is highly significant as weather emulators move toward $O(100)$ billion parameters, necessitating grounded principles for resource allocation.

Reference:
1. Zeng, Xinyue, et al. (2025). LENSLLM: Unveiling Fine-Tuning Dynamics for LLM Selection.
2. Pritzel, Alexander(2022). "GraphCast: Learning skillful medium-range global weather forecasting."

---

### Official Review · Reviewer_sKoe · 2026-02-22
**This paper studies how transformer based architecture scales with model, data, and compute budget for the weather forecasting domain. Using Swin Transformers, the authors systematically explore the trade-off model size, dataset size, and compute under isoFLOP constraints. The authors analyze how validation loss behaves when redistributing a fixed compute budget across different combinations of parameters and training data, and they identify compute-optimal regimes analogous to Chinchilla-style scaling in NLP. Because weather datasets are finite and strongly correlated, the study also examines the effects of multi-epoch training and learning rate scheduling, showing that prolonged constant learning rates followed by short cooldown phases can outperform full cosine decay schedules. Overall, the results suggest that weather models exhibit qualitatively similar compute–data–model trade-offs as foundation models in other domains, while highlighting domain-specific differences related to finite data regimes and optimization dynamics.**

**Rating:** 6
**Confidence:** 4

**Review:**

Strengths

1. This work explores neural scaling law in the weather forecasting domain using transformer-based architecture can be useful for architectural exploration in this domain.

2. The study carefully constructs isoFLOP curves to analyze trade-offs between model size, dataset size, and compute. The convex behavior and compute-optimal regimes are clearly illustrated and align qualitatively with known scaling-law behavior in other domains.

3. The comparison between cosine decay and constant learning rate with short cooldown is practically meaningful. The empirical finding that prolonged constant LR followed by a short refinement phase performs well is useful for practitioners working with moderate-scale scientific transformers.



Weaknesses

1. The explored parameter range (3M–450M) is relatively modest compared to prior large-scale scaling studies, which often span multiple orders of magnitude and include billion-parameter models. While sufficient to show structured isoFLOP trade-offs, the limited scale makes it unclear whether the observed trends would persist at larger model sizes or exhibit earlier saturation effects.

2. Pre-training a Swin Transformer and identifying compute-optimal points using established neural scaling methodologies offers limited methodological novelty. The core techniques (isoFLOP analysis and compute-optimal scaling) have been extensively studied in NLP and vision, and the paper primarily adapts these ideas to the weather domain rather than introducing new theoretical or algorithmic advances.

3. The paper refers to staged constant LR training as “continual learning,” this may be misleading. The approach is better described as multi-stage optimization rather than continual learning in the formal sense.

---

### Decision · Program_Chairs · 2026-03-03

Accept (Poster)